# Molecular Insights into Outer Dynein Arm Defects in Primary Ciliary Dyskinesia: Involvement of ZMYND10 and GRP78 [note 1]

**DOI:** 10.3390/cells14120916

**Published:** 2025-06-17

**Authors:** İlker Levent Erdem, Zeynep Bengisu Kaya, Pergin Atilla, Nagehan Emiralioğlu, Cemil Can Eylem, Emirhan Nemutlu, Uğur Özçelik, Halime Nayır Büyükşahin, Ayşenur Daniş, Elif Karakoç

**Affiliations:** 1Department of Histology and Embryology, Hacettepe University Faculty of Medicine, 06230 Ankara, Turkey; ilkerleventerdem@gmail.com (İ.L.E.); perginatilla@gmail.com (P.A.); 2Department of Neuroscience, Mayo Clinic, Jacksonville, FL 32224, USA; kaya.zeynep@mayo.edu; 3Department of Pediatric Pulmonology, Hacettepe University Faculty of Medicine, 06230 Ankara, Turkey; drnagehan@yahoo.com (N.E.); uozcelik@hacettepe.edu.tr (U.Ö.); hnayirbuyuksahin@gmail.com (H.N.B.); 4Department of Analytical Chemistry, Faculty of Pharmacy, Hacettepe University, 06230 Ankara, Turkey; cemilcaneylem@gmail.com (C.C.E.); emirhan.nemutlu@gmail.com (E.N.); 5Department of Pediatrics, University at Buffalo, Buffalo, NY 14203, USA; aysenurd@buffalo.edu

**Keywords:** PCD, ciliopathy, ODA, DNAH5, ZMYND10, GRP78, immunofluorescence, metabolomics, proteomics

## Abstract

Background: Primary ciliary dyskinesia (PCD) is a rare genetic disorder characterized by recurrent sinopulmonary infections due to motile cilia defects. The disease is genetically heterogeneous, with abnormalities in structural ciliary proteins. Zinc finger MYND-type containing 10 (ZMYND10) is essential for the assembly of outer dynein arms (ODA), with chaperones like Glucose-regulated protein 78 (GRP78) facilitating protein folding. This study investigates ZMYND10 and Dynein axonemal heavy chain 5 *(DNAH5*) mutations in individuals with PCD. Methods: Eight individuals aged 14–22 with clinical PCD symptoms and confirmed *DNAH5* mutations were included. We analyzed the correlation between *DNAH5* abnormalities and preassembly/chaperone proteins using immunofluorescence labeling. Nasal swabs were double-labeled (*DNAH5*–β-tubulin, β-tubulin–ZMYND10, β-tubulin–GRP78) and examined via fluorescence microscopy. Serum metabolomics and proteomics were also assessed. Results: The corrected total cell fluorescence (CTCF) levels of *DNAH5*, ZMYND10, and GRP78 were significantly different between PCD individuals and controls. Metabolomic analysis showed reduced valine, leucine, and isoleucine biosynthesis, with increased malate and triacylglycerol biosynthesis, malate-aspartate and glycerol phosphate shuttles, and arginine/proline metabolism, suggesting mitochondrial and ER stress. Conclusions: The altered expression of DNAH5, ZMYND10, and GRP78, along with metabolic shifts, points to a complex link between ciliary dysfunction and cellular stress in PCD. Further studies are needed to clarify the underlying mechanisms.

## 1. Introduction

Primary ciliary dyskinesia (PCD) (OMIM: 244400) is a rare, genetically heterogeneous disorder primarily caused by defects in motile cilia or flagella [1,2,3]. The most prevalent inheritance pattern observed is autosomal recessive, though instances of X-linked inheritance have been documented [4,5]. The estimated prevalence of this disease ranges from 1:10,000 to 1:20,000, with a higher incidence in regions like Turkey, where consanguineous marriages are more common [6].

The biogenesis and function of motile cilia are dependent on the presence of hundreds of proteins, as demonstrated by proteomic and genomic studies [7]. To date, mutations in more than 50 genes have been associated with PCD, underscoring its genetic heterogeneity [8,9]. However, the intracellular mechanisms linking protein synthesis to defective ciliogenesis remain unclear.

Motile cilia are specialized surface projections that are found predominantly in the respiratory tract and the male reproductive system [10,11,12]. Structurally, they exhibit a characteristic 9 + 2 axonemal configuration, consisting of nine peripheral microtubule doublets and two central microtubules [10,11,12]. These microtubules are linked by dynein arms, which can be categorized into two types: inner dynein arms (IDA) and outer dynein arms (ODA) [10,11,12]. Dynein axonemal heavy chain 5 (*DNAH5*) is the most frequently mutated gene in PCD, primarily causing outer dynein arm (ODA) defects that account for about 90% of cases [13]. *DNAH5* mutations represent 15–29% of the PCD cases in North America and Europe and 18% in China [14], and are responsible for up to 53% of all ODA defects [15].

The assembly of dynein motor complexes is initiated in the cytoplasm, where the synthesis of precursor proteins occurs, and they are subsequently transported to the ciliary axoneme via intraflagellar transport (IFT) [16,17,18]. This process necessitates appropriate folding and assembly, facilitated by a group of proteins known as axonemal dynein assembly factors (DNAAFs), in conjunction with molecular chaperones. DNAAF mutations cause combined IDA and ODA defects, underscoring their critical role in ciliogenesis [19,20,21,22,23].

One such DNAAF, Zinc finger MYND-type containing 10 (*ZMYND10*), encoded on chromosome 3p21, functions as a preassembly factor in the cytoplasm. A number of studies have identified a correlation between *ZMYND10* mutations and PCD; however, the precise mechanisms underlying this relationship remain to be fully elucidated [24,25,26].

Glucose-regulated protein 78 (GRP78), also known as BiP, is a key molecular chaperone that binds misfolded proteins and initiates endoplasmic reticulum-associated degradation (ERAD). GRP78 has been shown to regulate the unfolded protein response (UPR), a process which modulates protein translation and enhances proper protein folding under stress conditions [27].

Despite progress in understanding cilia’s structure and associated gene mutations, the intracellular mechanisms involving protein folding, chaperone function, and precursor transport remain poorly defined in the context of PCD. To address this gap, we investigated the relationship between the ciliary preassembly factor *ZMYND10* and the molecular chaperone GRP78 in individuals with confirmed *DNAH5* mutations. Our study aimed to elucidate the post-translational and stress-related regulatory pathways that may underlie dynein arm assembly defects.

Despite carrying identical *DNAH5* mutations, individuals demonstrated variable immunofluorescence labeling and divergent metabolomics and proteomics profiles, indicating distinct molecular phenotypes. These findings support the idea that post-translational regulation, cellular stress responses, or modifier pathways may shape phenotypic diversity in PCD patients.

## 2. Materials and Methods

### 2.1. Study Design

This study included 8 individuals from Turkey, who had been genetically diagnosed with PCD associated with *DNAH5* deficiency. The study population consisted of 8 participants aged between 14 and 22 years (3 males and 5 females) along with age- and sex-matched healthy controls.

All 8 individuals met the inclusion criteria and provided informed consent prior to participation.

The inclusion criteria required a confirmed genetic diagnosis of PCD due to *DNAH5* mutations and clinical stability. The exclusion criteria encompassed acute respiratory infections, chest trauma, coexisting respiratory or chronic conditions such as asthma or cardiovascular disease, and a lack of parental/individual consent [28].

### 2.2. Public and Patient Involvement Statement

The study protocol was approved by the Non-Interventional Clinical Research Ethics Committee of Hacettepe University. Written informed consent was obtained from all participants or their legal guardians.

### 2.3. Physical Examination

A comprehensive physical examination was conducted by the Department of Pediatric Pulmonology at Hacettepe University Faculty of Medicine. Individuals presenting with typical symptoms of PCD were evaluated by pediatric pulmonologists. Clinical assessments included screening for respiratory symptoms, chest auscultation, nasopharyngeal examination, measurement of nasal nitric oxide (nNO) through a NIOX-MINO (Circassia, Oxford, UK) device, and chest radiography.

### 2.4. Sample Collection

Nasal brushing samples, nasal swabs, and blood samples were collected by pediatric pulmonologists. Nasal brushing samples were obtained from the inferior turbinate and immediately placed into RPMI medium (R0883, Sigma-Aldrich, St. Louis, MO, USA). Nasal swabs were transferred to the Histology Laboratory for immunofluorescence (IF) analysis, while blood samples were sent to the Department of Analytical Chemistry for proteomic and metabolomic investigations. Healthy individuals without clinical suspicion of PCD were recruited as the control group.

### 2.5. High-Speed Video Microscopy (HSVM)

Suspended nasal epithelial cells were transferred onto glass slides and examined using high-speed video microscopy (HSVM). Ciliary beat patterns and amplitudes were recorded and analyzed accordingly [29].

### 2.6. Immunofluorescence (IF) Labeling

Nasal samples were dipped into the medium and smeared onto slides. The slides were air-dried for 24 h at room temperature (RT) and stored at −80 °C until further use [30,31]. Before staining, slides were equilibrated to RT for 2 h. After a 5 min wash with PBS, samples were fixed in 4% paraformaldehyde (PFA) for 15 min and permeabilized with 0.2% Triton X-100 for 10 min. Blocking was performed overnight at +4 °C using 1% skimmed milk. The slides were then incubated with primary antibodies overnight at +4 °C, followed by a PBS wash and a 30 min incubation with secondary antibodies at RT in the dark. Nuclei were stained with DAPI (D9542, Sigma, USA). A list of all antibodies used is provided in Appendix A.

At least three areas per sample were imaged using a Leica DM6B upright digital research microscope with LasX software (version 3.0.0.15697). The corrected total cell fluorescence (CTCF) was calculated for each image [30]. The fluorescence measurements were averaged for each sample, and the statistical significance was determined based on the mean values.

Corrected total cell fluorescence (CTCF) was calculated as follows: Integrated Density (IntDen value obtained by selecting All)–(Area of selected cell (Area value obtained by selecting All) X Mean fluorescence of background readings (Mean value obtained by selecting Oval) [30].

### 2.7. Proteomics and Metabolomics

Proteomic analyses were performed on serum samples using the shotgun technique at the Department of Analytical Chemistry, Hacettepe University, Faculty of Pharmacy. The obtained protein pellets were denatured by dissolving them in 100 mM ammonium bicarbonate (20% methanol) solution and adding DTT after removing the high amount of 12 proteins in the serum samples. After alkalizing with iodoacetamide, the proteins were broken down into their peptides by incubation with trypsin enzyme at 37 °C. At the end of the incubation period, the peptide-containing solutions were evaporated to dryness and analyzed by LC-qTOF-MS after dissolving in (*v*/*v*) acetonitrile containing 0.1% formic acid. The MS/MS data obtained as a result of the analysis, which can be extended as needed, were screened using the Maxquant software (version 1.16.17; Max Planck Institude of Biochemistry, Martinsried, Germany). The data were analyzed using a mass tolerance of 20 ppm, with carbamidomethylation on cysteine set as a fixed modification, and oxidation on methionine and N-terminal acetylation set as variable modifications. In the qualitative analysis of peptides, the FDR value was set to 0.01. The Human Uniprot database was used for the identification of proteins based on the determined peptides. A two-way *t*-test was applied to the program to examine whether the changes were significant or not. Proteins with significant differences were analyzed with the PANTHER program, and the role and pathways of proteins in serum and sodium during the ciliopathy process were clarified.

### 2.8. Statistical Analysis

All data were analyzed using Graph Pad Prism 8.4.2 software (San Diego, CA, USA). The normal distribution of the variables was analyzed by the Shapiro–Wilk test. The parametric data multiple comparisons were evaluated with one-way variance analysis (ANOVA) and the post-hoc Tukey test, and pairwise comparisons were evaluated using the *t*-test. For non-parametric data, Kruskal–Wallis and post-hoc Dunn tests were conducted and pairwise comparisons were evaluated using the Mann–Whitney U test. All analyses were plotted with a statistical significance degree of *p* < 0.05. Descriptive data were presented as the mean (standard deviation) for parametric variables, and as the median (interquartile range) for non-parametric variables.

## 3. Results

### 3.1. Clinical Features

Each of the eight individuals with the *DNAH5* mutation had unique symptoms. Individuals 1, 2, and 8 had rhinitis with a valid percent of 37.5% (one female and two males) and individuals 1, 2, 3, 5, and 7 had sinusitis with a valid percent of 62.5% (three females and two males). Individuals 2, 3, 4, 5, 6, and 7 had recurrent lung infection with a valid percent of 75% (four females and two males). Individuals 3, 4, and 5 had situs inversus with a valid percent of 37.5% (three females) (Figure 1). Individuals 2, 3, 4, and 8 had consanguinity with a valid percent of 50% (three females and one male). Individuals 1, 2, 3, 4, 5, 6, and 7 had bronchiectasis with a valid percent of 87.5% (five females and two males). The consanguinity ratio was a valid percent of 50% (three females and only one male in four individuals). Individuals 5 and 8 had a hearing deficit with a valid percent of 25% (one female and one male). All individuals had low nNO levels with a cut-off value (11.5 ppb) [30] with a valid percent of 100% (five females and three males) (Table 1).

### 3.2. HSVM Findings

According to the high-speed video microscopy results, individuals 1, 2, 4, 5, and 6 displayed almost immotile cilia with only minimal residual ciliary movements, and individuals 3 and 7 had slightly reduced amplitude (Table 2).

### 3.3. Genetic Characteristics of Individuals

Despite all individuals carrying mutations in the *DNAH5* gene, variations were observed in mutation type, coding region location, and zygosity (Table 2 and Appendix A). Homozygous variants were found in individuals 2, 3, 4, 6, 7, and 8, while individuals 1 and 5 carried heterozygous variants (Table 2 and Appendix A).

Individual 1, who presented with bronchiectasis and classical upper airway symptoms, carried a monoallelic missense variant in *DNAH5* (c.11740G>A; p.Glu3914Lys). Individual 2, a descendant of a consanguineous family, had classical upper airway symptoms, bronchiectasis, and low nasal NO, and carried biallelic nonsense variants (c.13486C>T; p.Arg4496Ter). Individuals 3 and 4, also from consanguineous families and both with bronchiectasis, situs inversus, recurrent lung infection, carried biallelic missense variants: Individual 3 carried c.7615T>C (p.Trp2539Arg), and Individual 4 carried c.8897C>T (p.Thr2966Met). Individual 3 had also low nasal NO. Individual 5 exhibited classical upper airway symptoms, recurrent lung infection, bronchiectasis, hearing defects, and low nasal NO, and carried a monoallelic nonsense variant (c.5747G>A; p.Trp1916Ter). Individuals 6 and 8 both harbored biallelic nonsense variants: Individual 6 (c.2710G>T; p.Glu904Ter) had bronchiectasis, recurrent lung infection, and low nasal NO, and Individual 8 (c.9502C>T; p.Arg3168Ter) was from a consanguineous family and exhibited classical upper airway symptoms, hearing defects, and low nasal NO. Lastly, Individual 7, also with classical upper airway symptoms, low nasal NO, and bronchiectasis, carried a biallelic missense variant (c.2368G>C; p.Ala790Pro).

### 3.4. Immunofluorescence (IF) Analysis

#### 3.4.1. DNAH5

In the control samples, both *DNAH5* and β-tubulin were continuously expressed along the entire proximal-to-distal axis of the cilium (Figure 2A). In contrast, *DNAH5*-mutant individuals exhibited variable DNAH5 labeling patterns (Figure 2B–I). Individuals 1, 2, 3, 4, and 6 displayed no detectable *DNAH5* expression along the ciliary shaft. Individual 5 exhibited reduced *DNAH5* expression limited to the proximal cilium, while Individual 7 demonstrated expression predominantly in the distal region. Individual 8 exhibited *DNAH5* expression extending along the entire cilium.

Quantitative analysis of DNAH5 fluorescence intensity (CTCF) showed a statistically significant reduction in individuals 1, 2, 3, 4, and 6 compared to the controls (* *p* < 0.05, Mann–Whitney U test). No significant differences were detected between the controls and individuals 5, 7, and 8 (Figure 2J).

#### 3.4.2. ZMYND10

β-Tubulin localized along the cilia, and *ZMYND10* was distributed throughout the control samples’ cytoplasm (Figure 3A). In contrast, *DNAH5*-mutant individuals exhibited a normal β-Tubulin labeling pattern, while ZMYND10 expression was markedly reduced or absent (Figure 3B–I).

Quantitative analysis of ZMYND10 expression using corrected total cell fluorescence (CTCF) demonstrated a significant difference between control and *DNAH5*-mutant samples (* *p* < 0.05, Mann–Whitney U test) (Figure 3J).

#### 3.4.3. GRP78

In the control samples, β-Tubulin localized along the cilia, and GRP78 was consistently expressed in the cytoplasm following a pattern similar to ZMYND10 (Figure 4A). In contrast, *DNAH5*-mutant individuals exhibited normal β-Tubulin staining but reduced or absent GRP78 expression (Figure 4B–I).

Quantitative analysis of GRP78 expression using corrected total cell fluorescence (CTCF) revealed a significant difference between control and *DNAH5*-mutant samples (* *p* < 0.05, *t*-test) (Figure 4J).

### 3.5. Proteomics and Metabolomics Results

Proteomic analysis of serum samples from eight individuals suspected of PCD revealed both shared and individual-specific alterations in their protein expression profiles compared to healthy controls. Notably, individuals 1 and 4 showed significant upregulation of Zinc Finger Protein 415 (ZNF415), with the expression levels in Individual 4 slightly exceeding those in Individual 1. In Individual 4, Small Subunit Processome Component (UTP18) was markedly downregulated, while other proteins such as Signal Transducer and Activator of Transcription 6 (STAT6), Zinc Finger RANBP2-Type Containing 3 (ZRANB3), and NTPase KAP Family P-Loop Domain Containing 1 (NKPD1) were notably upregulated. Similarly, Individual 1 showed elevated levels of Family with Sequence Similarity 196 Member A (FAM196A), ZRANB3, Coiled-Coil Domain Containing 173 (CCDC173), STAT6, Proteasome Activator Subunit 4 (PSME4), Glycogen Synthase Kinase 3 Beta (GSK3B), and Calreticulin (CALR). Conversely, several proteins were downregulated in Individual 1, including ATP-Binding Cassette Subfamily A Member 13 (ABCA13), Ubiquinol-Cytochrome c Reductase Core Protein 2 (UQCRC2), Coiled-Coil Domain Containing 71 Like (CCDC71L), Zinc Finger Protein 521 (ZNF521), and Serpin Family C Member 1 (SERPINC1).

Individual 2 displayed elevated serum levels of CALR, Ceruloplasmin (CP), Leucine-Rich Alpha-2-Glycoprotein 1 (LRG1), Uracil-DNA Glycosylase (UNG), Complement Factor H (CFH), and NKPD1, while levels of FAM196A, ZNF521, ABCA13, and UQCRC2 were decreased. Individual 3 exhibited a marked reduction in Uridine Phosphorylase 1 (UPP1) and UTP18, along with increased levels of NKPD1, CFH, and CCDC173. Similarly, Individual 8 also showed reduced UPP1 and UTP18 levels, suggesting a potential shared molecular signature. However, Individual 8 had a distinct profile with significantly increased Ectonucleoside Triphosphate Diphosphohydrolase 1 (ENTPD1) levels and downregulation of LRG1, CALR, ZNF521, ZRANB3, and STAT6.

In Individual 5, elevated expression of CALR, SEC13 Homolog (SEC13), STAT6, and LRG1 was observed, while cation channel sperm-associated protein (CATSPERG), CCDC173, FAM196A, and ENTPD1 were downregulated. Individual 6 displayed notable downregulation of SERPINC1, CCDC71L, ABCA13, ZNF521, ZNF415, and GSK3B, whereas PSME4, SEC13, and UNG were elevated. Individual 7 exhibited reduced levels of CATSBERG, UPP1, CCDC71, and ABCA13, with concurrent upregulation of Mannosidase Beta Like (MANBAL), SPANXA2 Opposite Strand Transcript 1 (SPANXA2-OT1), and ZNF415 (Figure 5).

Metabolomics analysis revealed that in individuals suspected of having PCD, pathways such as valine, leucine, and isoleucine biosynthesis; arginine biosynthesis; nicotinamide and histidine biosynthesis; beta-alanine metabolism; pantothenate and CoA biosynthesis; pyruvate metabolism; gluconeogenesis and glycolysis; and glutathione metabolism were upregulated compared to the controls. In contrast, lipid biosynthesis and tryptophan metabolism were downregulated (Figure 6).

## 4. Discussion

PCD is a rare genetic disorder characterized by impaired mucociliary clearance, typically resulting from defects in motile cilia structure and function. Among the genes implicated, *DNAH5* is responsible for encoding a core component of the ODA, and it is one of the most frequently mutated genes in *DNAH5*-mutant individuals diagnosed with PCD [13,14,29]. In this study, we focused on individuals from Turkey with confirmed mostly biallelic DNAH5 mutations, and the molecular profile of ODA dysfunction was examined through a combination of immunofluorescence microscopy, proteomic/metabolomic analyses, and clinical characterization.

### 4.1. Clinical Findings

The clinical profiles of the eight subjects with mostly biallelic *DNAH5* mutations in the present study largely reflect classical presentations of PCD, including chronic upper and lower airway symptoms, and recurrent infections [32]. In accordance with the findings of this research, the majority of cases exhibited respiratory symptoms, including recurrent airway infections, sinusitis, bronchiectasis, and hearing deficits, consistent with the established function of *DNAH5* in ODA formation and motile ciliary function [29,33].

Despite the shared genetic etiology, heterogeneity in the clinical findings was observed. This variability may be indicative of the influence of modifier genes, environmental exposures, or differences in ciliary ultrastructure due to the nature or location of specific *DNAH5* variants. Three *DNAH5* mutant individuals demonstrated situs inversus, indicative of impaired nodal cilia function [34]. Our findings are consistent with these observations.

Furthermore, a significant decrease in nNO levels was observed in all participants. Genetics, IF microscopic findings, and HSVM demonstrating severely dyskinetic ciliary beating patterns were present, confirming the diagnosis [32]. Wang et al. identified novel compound heterozygous mutations of *DNAH5* [33] and Dalal A. Al-Mutairi et al. showed novel pathogenic variants of *DNAH5* associated with the clinical and genetic spectra of PCD in an Arab population [35]. Our results were consistent with their studies as well.

### 4.2. DNAH5 Immunolabeling Showed Variety in DNAH5-Mutant Individuals

Individuals with *DNAH5* mutations exhibited variable and often reduced ciliary labeling. Individuals 1, 2, 3, 4, and 6 lacked detectable DNAH5 expression, consistent with severe loss-of-function variants [35]. Individuals 5 and 7 showed regionally restricted expression (proximal or distal), while Individual 8 retained near-complete labeling, suggesting residual protein function. Individual 5, heterozygous for a nonsense mutation (c.5747G>A; p.Trp1916*), exhibited proximal axonemal labeling, which may indicate reduced protein expression or impaired axonemal transport. These results highlight the complexity of correlating the *DNAH5* genotype with protein localization patterns and reinforce the need for integrative analysis including genetic, immunofluorescent, and ultrastructural data [36]. Individual 7, with a homozygous missense mutation (c.2368G>C; p.Ala790Pro), showed labeling predominantly at the distal portion of the ciliary axoneme. This suggests partial protein production and correct distal targeting, consistent with previous reports that certain missense variants may allow residual dynein arm assembly and localization [37]. In contrast, Individual 8, who carried a homozygous nonsense mutation (c.9502C>T; p.Arg3168*), displayed labeling along the entire axoneme, a finding that may be explained by the late position of the premature stop codon, potentially allowing upstream translation and antibody recognition.

These findings underscore the phenotypic heterogeneity of *DNAH5*-related PCD and support the utility of immunofluorescence in assessing mutation impact [38]. The observed differences in DNAH5 distribution emphasize the importance of the molecular mechanisms underlying disease variability.

### 4.3. ZMYND10 Expression Is Altered in DNAH5-Related PCD

ZMYND10 is a key factor in the cytoplasmic preassembly of axonemal dynein arms, where it forms a co-chaperone complex with HSP90 and FKBP8 to help stabilize ODA components before they are transported to the cilia [39]. In the present study, ZMYND10 expression was reduced in the nasal epithelial cells of all eight individuals with *DNAH5* mutations. These findings suggest that mutations in *DNAH5* might be the result of the disruption of the ZMYND10-associated chaperone network, potentially due to stalled preassembly or the degradation of unincorporated subunits. This supports the idea that PCD is not only a structural defect in the axoneme, but also a disorder of proteome homeostasis. In line with this, Mali et al. reported that *Zmynd10*-knockout mice exhibited significantly reduced levels of multiple ODA proteins, including DNAH5 [39]. Our results reflect a similar pattern, with reduced ZMYND10 expression observed in all *DNAH5*-mutant individuals.

### 4.4. GRP78 Downregulation Indicates ER Stress in PCD Epithelium

A novel finding of our study is the downregulation of GRP78 (BiP) in *DNAH5*-mutated individuals, as evidenced by both immunofluorescence and systemic proteomic profiling. GRP78 is a master regulator of the UPR, which is activated upon the accumulation of misfolded proteins in the endoplasmic reticulum (ER) [40]. Its decreased expression in PCD nasal epithelial cells suggests that chronic ER stress may be a consequence of defective dynein assembly and turnover, further impairing ciliary biogenesis and epithelial function. This ER stress response may be indicative of a compensatory mechanism or an exacerbating factor in PCD pathophysiology.

### 4.5. Serum Proteomics and Metabolomics Support Systemic Stress Pathways

In addition to ciliary protein labeling pattern changes, our serum proteomic and metabolomic analyses revealed broader molecular alterations in pathways linked to mitochondrial function, oxidative phosphorylation, amino acid metabolism, and ER-mitochondria crosstalk. Among all *DNAH5*-mutant individuals’ serum samples, inflammation/stress (STAT6, CALR), ER stress (CALR), proteostasis (PSME4), redox/metabolism (GSK3B, CFH, UNG), and energy production pathways (glycolysis/gluconeogenesis) were the most frequently affected pathways in proteomic and metabolomic profiles. In individuals 1, 2, 3, 4, and 6, multiple proteins related to ER stress, immune response, mitochondrial dysfunction, and redox control mechanisms were dysregulated. For instance, in individuals 1, 2, and 5, elevated CALR expression indicated an ER stress response to misfolded or unassembled DNAH5 protein. And Individual 1 had increased CALR, GSK3B, and STAT6 protein levels, suggesting that ER stress and inflammation may be secondary consequences or compensatory to the loss of DNAH5 immunolabeling and mutation. Individual 4’s increased ZNF415 and decreased UTP18 proteins might be due to altered transcription and ribosome biogenesis due to ciliary assembly failure. In Individual 5, increased CALR and STAT6 protein levels might be the reason for proximal DNAH5 labeling and moderate ER stress signature. Individual 7’s increased ZNF415 and novel transcript SPANXA2-0T1 levels and distal DNAH5 labeling could be explained by differential cellular responses. Despite normal Ciliary labeling, the *DNAH5*-mutated Individual 8 had increased ENTPD1 and decreased UPP1, UTP18, LRG1, CALR, and STAT6 protein levels, which might indicate a compensated or post-repair phenotype, mosaicism, or non-ciliary systemic abnormalities despite the DNAH5 label being present [41].

These findings are consistent with recent studies suggesting that PCD may have systemic metabolic signatures, potentially reflecting chronic inflammation, oxidative stress, or adaptive cellular responses to ciliary dysfunction [42].

### 4.6. Study Limitations

This study is limited by its small sample size and lack of functional or mechanistic validation. Additionally, while immunofluorescence offers valuable localization data, it does not quantify protein abundance with the same precision as Western blotting. Future studies incorporating quantitative proteomics, RNA sequencing, and rescue models (e.g., CRISPR-based correction or protein overexpression) would help to confirm the mechanistic link between *DNAH5* mutations, *ZMYND10* misregulation, and ER stress responses [43].

## 5. Conclusions

Our findings demonstrate that *DNAH5* mutations in PCD disrupt not only ciliary structure, but also cytoplasmic dynein preassembly, as evidenced by uniformly reduced ZMYND10 expression across all affected individuals. This likely reflects the impaired chaperone-mediated stabilization of ODA components, supporting the idea that PCD involves proteostasis failure. The concurrent downregulation of GRP78 (BiP), both locally and systemically, further indicates chronic ER stress due to misfolded or unassembled dynein proteins. Proteomic and metabolomic alterations—including the downregulation of lipid and tryptophan metabolism and the upregulation of stress-related pathways—suggest a compensatory cellular response to protein folding stress. Importantly, individuals with absent DNAH5 labeling exhibited the most severe molecular disturbances, reinforcing the link between dynein arm deficiency and systemic cellular stress.

The consistent immunofluorescence patterns of DNAH5 loss, altered ZMYND10 localization, and GRP78 may offer a complementary diagnostic approach, particularly in centers with limited access to genetic sequencing. Furthermore, these markers provide insight into the interplay between ciliary structural proteins, chaperone networks, and stress signaling pathways, which may be leveraged for future therapeutic development. Notably, the utilization of targeted immunofluorescence labelling for DNAH5, ZMYND10, and GRP78 has the potential to enhance the diagnostic algorithm, especially in cases where genetic testing has produced inconclusive results or is unavailable.

## Figures and Tables

**Figure 1 cells-14-00916-f001:**
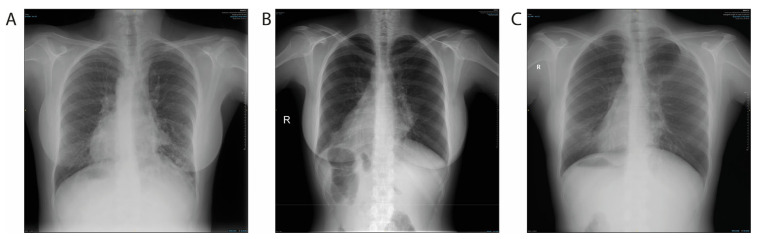
Posterior–anterior chest X-ray images of DNAH5-mutant individuals. (**A**) Individual 3, (**B**) Individual 4, and (**C**) Individual 5 had widespread bronchiectasis and dextrocardia.

**Figure 2 cells-14-00916-f002:**
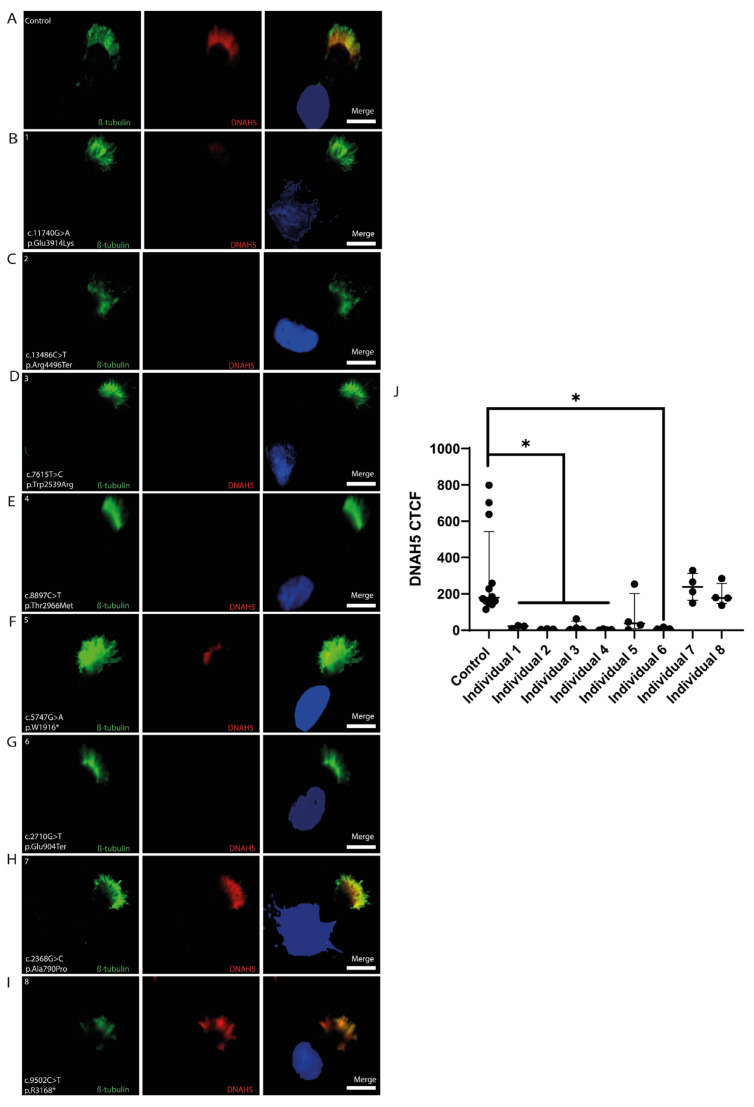
DNAH5 and β-Tubulin labeling in the respiratory epithelium of the control and PCD-affected individuals (1-8) harboring *DNAH5* pathogenic variants. (**A**) Cilia from a healthy control showing double immunofluorescence labeling with antibodies against DNAH5 (red) and β-tubulin (green) along the entire ciliary length. (**B**–**E**,**G**) The DNAH5 signal is undetectable in the respiratory cilia of individuals with *DNAH5* mutations. (**F**) DNAH5 shows proximal localization in *DNAH5*-mutant cilia. (**H**) DNAH5 displays distal localization in *DNAH5*-mutant cilia. (**I**) DNAH5 and β-tubulin are colocalized along the ciliary length of *DNAH5*-mutant cilia. Nuclei were stained in DAPI (blue). Scale bars represent 10 μm. (**J**) Quantitative analysis of DNAH5 fluorescence intensity (CTCF) showed a statistically significant reduction in individuals 1, 2, 3, 4, and 6 compared to the controls (* *p* < 0.05, Mann–Whitney U test). No significant differences were detected between the controls and individuals 5, 7, and 8. Data were plotted as the median (interquartile range).

**Figure 3 cells-14-00916-f003:**
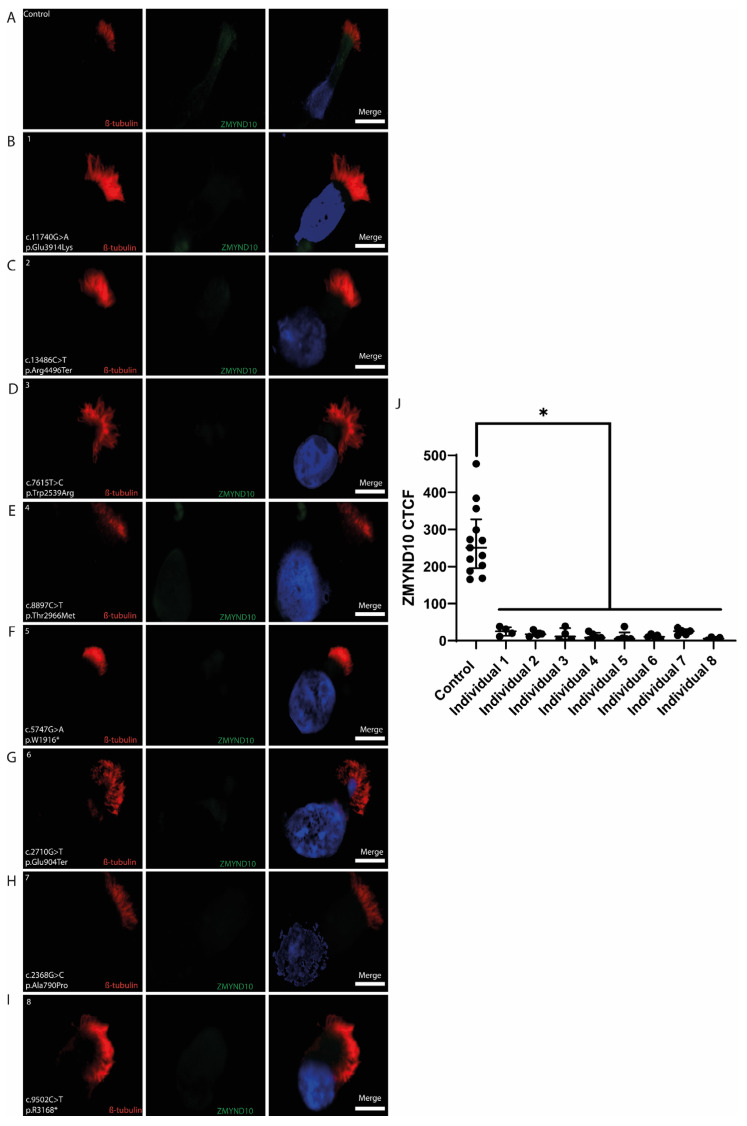
Immunolabeling of the ciliary preassembly factor ZMYND10 and β-Tubulin in the respiratory ciliary axonemes of the control and PCD-affected individuals (1-8) with pathogenic variants of *DNAH5.* (**A**) Cilia from a healthy control showing double immunofluorescence labeling with β-tubulin (red) along the entire ciliary length and ZMYND10 (green) distribution throughout the cytoplasm. (**B**–**I**) DNAH5-mutant individuals exhibited a normal β-Tubulin labeling pattern, while ZMYND10 expression was markedly reduced or absent. Nuclei were stained in DAPI (blue). Scale bars represent 10 μm. (**J**) Quantitative analysis of ZMYND10 expression using corrected total cell fluorescence (CTCF) demonstrated a significant difference between control and DNAH5-mutant samples (* *p* < 0.05, Mann–Whitney U test). Data were plotted as the median (interquartile range).

**Figure 4 cells-14-00916-f004:**
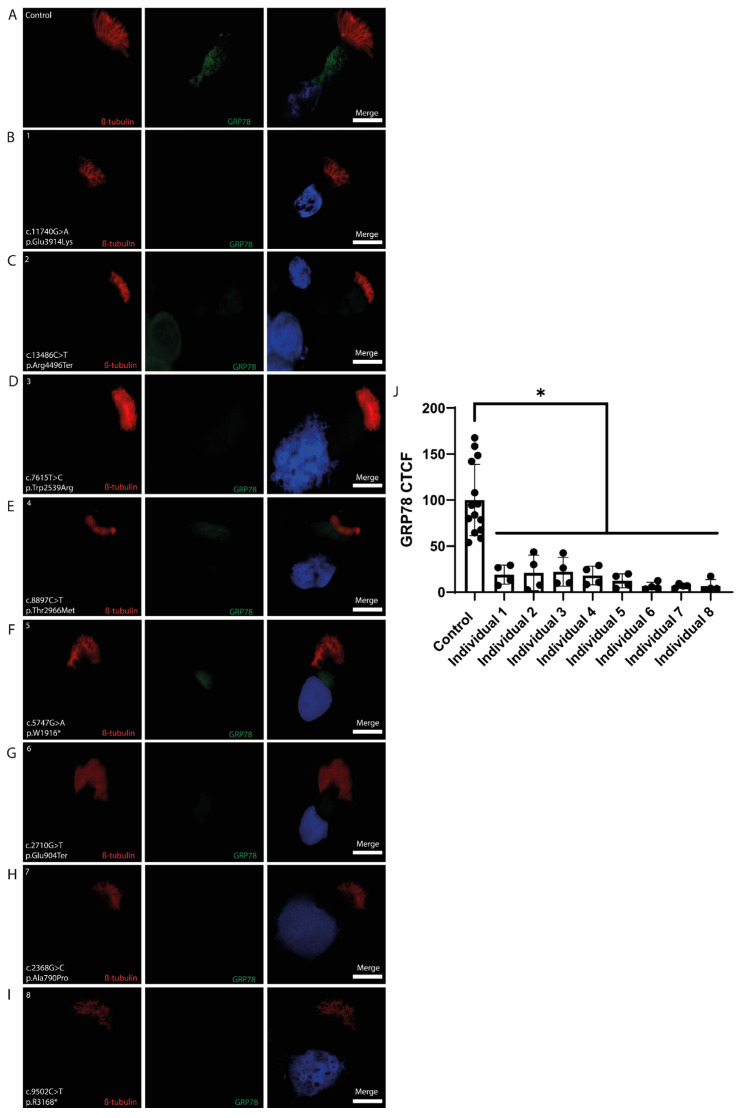
Immunolabeling of the chaperone protein GRP78 and β-Tubulin in the respiratory ciliary axonemes of the control and PCD-affected individuals (1-8) with pathogenic variants of *DNAH5*. (**A**) Cilia from a healthy control showing double immunofluorescence labeling with β-tubulin (red) along the entire ciliary length and GRP78 (green) distribution throughout the cytoplasm. (**B**–**I**) *DNAH5*-mutant individuals exhibited a normal β-Tubulin labeling pattern, while GRP78 expression was markedly reduced or absent. Nuclei were stained in DAPI (blue). Scale bars represent 10 μm. (**J**) Quantitative analysis of GRP78 expression using corrected total cell fluorescence (CTCF) revealed a significant difference between control and DNAH5-mutant samples (* *p* < 0.05, *t*-test). Data were plotted as the mean (standard deviation).

**Figure 5 cells-14-00916-f005:**
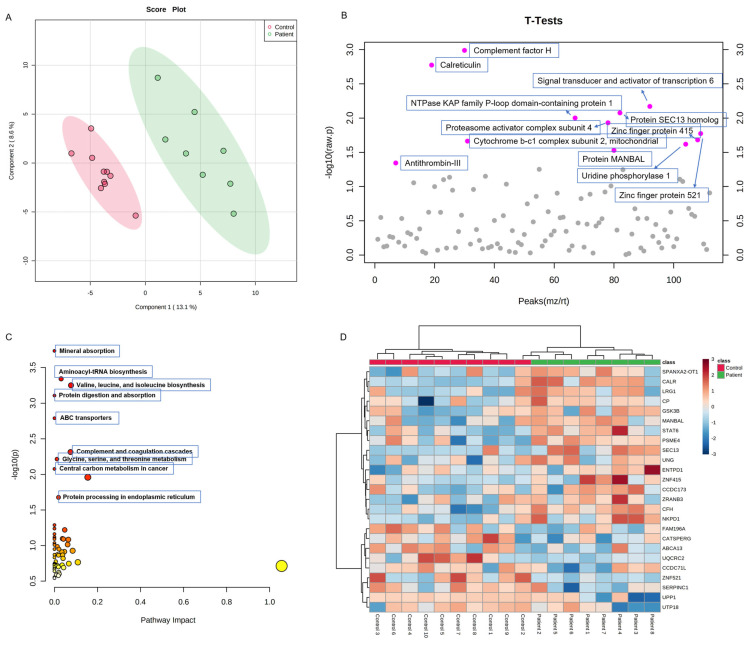
(**A**) A PCA score plot based on the proteomic profiles of the control and patient groups. (**B**) A univariate statistical analysis of the proteomic data using *t*-tests. The plot displays the distribution of peaks (*m*/*z* or retention time) against the statistical significance (–log_10_ *p*-value) of differences between the control and patient groups. The proteins with the most significant differences (*p* < 0.05) are highlighted and annotated. (**C**) A joint pathway analysis of the significantly altered (*p* < 0.05) proteins and metabolites between the control and *DNAH5*-mutant individual groups. The bubble plot displays metabolic pathways based on their significance (–log_10_(*p*) value, y-axis) and pathway impact (topological importance, x-axis). (**D**) A heatmap of the differentially expressed proteins between the control and patient groups.

**Figure 6 cells-14-00916-f006:**
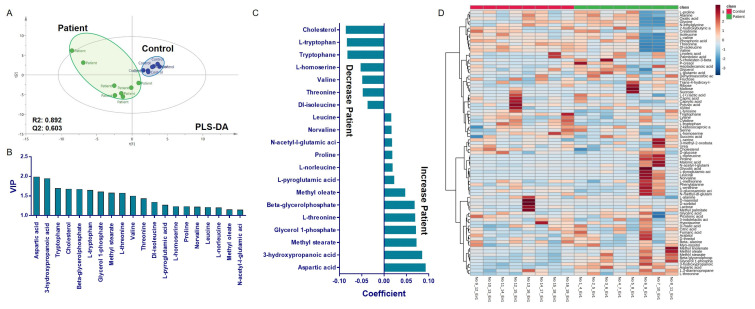
Multivariate statistical analysis and metabolite selection between the patient and control groups. (**A**) A PLS-DA score plot showing clear separation between the patient and control groups, indicating distinct metabolic profiles (R^2^ = 0.892, Q^2^ = 0.603). (**B**) The VIP scores of the top discriminatory metabolites identified in the PLS-DA model. (**C**) A coefficient plot of the PLS-DA model indicating the direction and magnitude of each metabolite’s contribution to the group’s classification; positive coefficients indicate higher abundance in the patients, whereas negative coefficients indicate higher abundance in the controls. (**D**) A heatmap of the differentially abundant metabolites between the control and patient groups.

**Table 1 cells-14-00916-t001:** Descriptive statistics of individuals.

Clinical Statistics			Valid Percent
Gender Female Male	n 5 3		62.5 37.5
Age median (min–max)		17.5 (14–22)	
nNO median (min–max)		7.5 (5–10)	
Consanguinity, n	4 (3 females; 1 male)		50
Sinusitis, n	5 (3 females; 2 males)		62.5
Bronchiectasis, n	7 (5 females; 2 males)		87.5
Recurrent lung infection, n	6 (4 females; 2 males)		75
Hearing defect, n	2 (1 female; 1 male)		25
Situs inversus, n	3 (3 females)		37.5
Rhinitis, n	3 (1 female and 2 males)		37.5

**Table 2 cells-14-00916-t002:** Clinical, genetic, and demographic findings of PCD-affected individuals.

Individual ID	Sex	Age	Symptoms	Bronchiectasis	Situs Inversus	Consanguinity	Lobectomy	Nasal NO (ppb)	HSVM	Genetic
1	M	21 y	Rhinitis, sinusitis	yes	no	no	no	5 ppb	Almost immotile cilia, only minimal residual ciliary movements	DNAH5 het: c.11740G>A; p.Glu3914Lys
2	F	17 y	Recurrent lung infection, rhinitis, recurrent sinusitis	yes	no	yes	no	6 ppb	Almost immotile cilia, only minimal residual ciliary movements	DNAH5 hom: c.13486C>T; p.Arg4496Ter
3	F	22 y	Recurrent lung infection, sinusitis	yes	yes	yes	no	8 ppb	Slightly reduced amplitude	DNAH5 hom: c.7615T>C; p.Trp2539Arg
4	F	18 y	Recurrent lung infection	yes	yes	yes	no	6 ppb	Almost immotile cilia, only minimal residual ciliary movements	DNAH5 hom: c.8897C>T; p.Thr2966Met
5	F	19 y	Recurrent lung infection, sinusitis, hearing defect	yes	yes	no	no	10 ppb	Almost immotile cilia, only minimal residual ciliary movements	DNAH5 het: c.5747G>A; p.W1916 *
6	M	16 y	Recurrent lung infection	yes	no	no	yes	7 ppb	Almost immotile cilia, only minimal residual ciliary movements	DNAH5 hom: c.2710G>T; p.Glu904Ter
7	F	16 y	Recurrent lung infection, sinusitis,	yes	no	no	yes	9 ppb	Slightly reduced amplitude	DNAH5 hom: c.2368G>C; p.Ala790Pro
8	M	14 y	Rhinitis, hearing defect	no	no	yes	no	8 ppb	NA	DNAH5 hom: c.9502C>T; p.R3168 *

F: female; M: male; y: years; ppb: parts per billion; NO: Nitric oxide; HSVM: high-speed video microscopy; NA: not available; *: premature stop codon.

## Data Availability

The data presented in this study are available on request from the corresponding author.

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
