# Peer review of "Molecular Insights into Outer Dynein Arm Defects in Primary Ciliary Dyskinesia: Involvement of ZMYND10 and GRP78†"

_cells, 2025, doi:10.3390/cells14120916_

Round 1

Reviewer 1 Report

Comments and Suggestions for Authors

Erdem et al. investigated how intracellular activities, such as protein synthesis and protein folding, are involved in axoneme dynein arm assembly and phenotype diversity in PCD. By analyzing clinical individuals with PCD symptoms, they demonstrated that DNAH5 mutations in PCD not only disrupt ciliary structure but also affect cytoplasmic dynein preassembly. Furthermore, they performed proteomic and metabolomic analyses of the serum samples from these PCD patients and found that the altered expression of axonemal dynein assembly factor ZMYND10 and chaperone GRP78. As they stated in the study limitations section, this study lacks functional or mechanistic validation. The images in the figures are of low quality and do not well support their conclusion. My main concerns regarding this manuscript are as follows.

  1. The representative images in Figures 3 and 4 do not align with the corresponding statistical results, which indicate that the fluorescence intensity in control samples is significantly higher than that in the mutant samples. However, it is hard to observe any obvious green fluorescence in the control samples.
  2. To compare fluorescence intensity between different samples, all images must be obtained under the same microscope settings. The authors should explain what caused the significant difference in the fluorescence intensity of the cell nuclei in Figures 2,3, and 4.
  3. The authors state that “A key novel finding of our study is the upregulation of GRP78 (BiP) in DNAH5-mutated individuals” in the discussion section. Based on my understanding, they attempt to conclude that the expression of GRP78 was decreased in individuals with DNAH5 mutations, although the images they provided do not clearly show this difference. The authors need to clarify how this statement was raised in the discussion.
  4. All the images throughout the manuscript should be reviewed and presented in high resolution to ensure that all the information in the images is clearly visible and accessible to readers.

Minor points

  1. The format of references should be checked.
  2. The letter annotations in the figures are inconsistent; some have brackets, whereas others do not.

Reviewer 2 Report

Comments and Suggestions for Authors

I read this manuscript with interest, as it explores the relationship between ZMYND10, a ciliary pre-assembly factor, and the molecular chaperone GRP78, using immunofluorescence, proteomics, and metabolomics approaches."

The authors examined 8 cases of PCD with mutations in DNAH5, 6 of which were homozygous and 2 heterozygous.  They conclude that in the nasal epithelial cells of these individuals, ZMYND10 expression is reduced and GRP78 is upregulated.

Overall, the manuscript is well-written and easy to understand, and the bibliography is complete and up to date.

I have only a few brief comments:

  1. In the introduction, the sentence reporting the prevalence percentages of DNAH5 mutations is, in my opinion, unclear and somewhat confusing. I suggest simplifying the sentence and enriching it with additional references.
  2. Although I am not a geneticist, it is my understanding that the panels commonly used for the genetic analysis of mutations in PCD cases also include ZMYND10. Could the Authors clarify whether a genetic analysis of this gene was performed on blood samples, and if so, what results were observed?
  3. Are there any available data on the ciliary ultrastructure of these patients?
  4. Supplementary video 1 showing ciliary motility in a control case is too short. I would suggest replacing it with a video of similar duration to those of the DNAH5 mutation cases.
  5. Could the study of ZMYND10 and GRP78 by immunofluorescence have diagnostic potential for PCD even in the absence of DNAH5 mutations?

Round 2

Reviewer 1 Report

Comments and Suggestions for Authors

The authors have adequately addressed the raised questions. I have no additional comments.